# Transcriptional Trajectories in Mouse Limb Buds Reveal the Transition from Anterior-Posterior to Proximal-Distal Patterning at Early Limb Bud Stage

**DOI:** 10.3390/jdb8040031

**Published:** 2020-12-07

**Authors:** Ines Desanlis, Rachel Paul, Marie Kmita

**Affiliations:** 1Genetics and Development Research Unit, Institut de Recherches Cliniques de Montréal, Montreal, QC H2W 1R7, Canada; Ines.Desanlis@ircm.qc.ca (I.D.); Racheal.Paul@ircm.qc.ca (R.P.); 2Département de Médecine, Université de Montréal, Montreal, QC H3T 1J4, Canada; 3Department of Experimental Medicine, McGill University, Montreal, QC H4A 3J1, Canada

**Keywords:** limb patterning, single-cell RNA-seq, *Hox* genes

## Abstract

Limb patterning relies in large part on the function of the *Hox* family of developmental genes. While the differential expression of *Hox* genes shifts from the anterior–posterior (A–P) to the proximal–distal (P–D) axis around embryonic day 11 (E11), whether this shift coincides with a more global change of A–P to P–D patterning program remains unclear. By performing and analyzing the transcriptome of the developing limb bud from E10.5 to E12.5, at single-cell resolution, we have uncovered transcriptional trajectories that revealed a general switch from A–P to P–D genetic program between E10.5 and E11.5. Interestingly, all the transcriptional trajectories at E10.5 end with cells expressing either proximal or distal markers suggesting a progressive acquisition of P–D identity. Moreover, we identified three categories of genes expressed in the distal limb mesenchyme characterized by distinct temporal expression dynamics. Among these are *Hoxa13* and *Hoxd13* (*Hox13* hereafter), which start to be expressed around E10.5, and importantly the binding of the HOX13 factors was observed within or in the neighborhood of several of the distal limb genes. Our data are consistent with previous evidence suggesting that the transition from the early/proximal to the late/distal transcriptome of the limb mesenchyme largely relies on HOX13 function. Based on these results and the evidence that HOX13 factors restrict *Hoxa11* expression to the proximal limb, in progenitor cells of the zeugopod, we propose that HOX13 act as a key determinant of P–D patterning.

## 1. Introduction

Limb development in mouse initiates around E9.5, when the forelimb bud emerges from the lateral plate mesoderm. At this stage, undifferentiated mesenchymal cells receive signals from a group of ectodermal cells at the distal tip of the limb bud, referred to as the Apical Ectodermal Ridge (AER). The AER, located at the boundary between the dorsal and ventral ectoderm, expresses Fibroblast Growth Factors (Fgfs), notably *Fgf8*, whose function promotes limb bud growth and provides distal positional information to the underlying mesenchyme [1]. Concomitantly, *Meis1/2* in the most proximal domain of the developing limb bud construes FGF signaling and contributes to establishing proximal positional information [2]. In parallel, cells at the distal-posterior margin of the limb bud express Sonic Hedgehog (*Shh*), establishing a gradient of SHH signaling along the anterior–posterior (A–P) axis, which underlies the A–P patterning of the developing limb [3]. Genes whose transcriptional control is influenced by FGFs and SHH signaling include the *Hox* family of developmental genes, which contribute to the limb patterning along both the A–P and P–D axes (reviewed in, e.g., [4]). *Hox* genes belonging to the *HoxA* and *HoxD* clusters play a major role in limb development, as illustrated by the severe limb truncation resulting from the conditional inactivation of both gene clusters [5]. These genes are expressed in two phases. During the first phase, at early limb bud stages, *HoxA/D* genes are sequentially activated in time, from group 1, at one end of each cluster, to group 13 genes, located at the opposite end. During this first phase, *Hox* genes are expressed in a nested fashion along the A–P axis, with the late activated genes being posteriorly restricted [6]. The second phase of expression starts around E11.5, with *HoxA/D* genes from group 9 to 13 being differentially expressed along the P–D axis [4]. The transcriptional control of *HoxA/D* genes relies on remote cis-regulatory elements, located on both sides of each *Hox* cluster, forming the early/proximal and late/distal regulatory landscapes respectively [7,8]. Elegant studies have shown that the switch of activity from one landscape (downstream of the group 1 genes) to the other regulatory landscape (upstream of the group 13 genes) is associated with the differential expression of *Hox* genes along the P–D axis and thereby the establishment of the three limb segments along the P–D axis [9,10]. While there is a clear shift in *Hox* regulation around E11.5, whether it coincides with a more global change of A–P to P–D patterning program remains unclear.

In this work, we have analyzed the complete transcriptome of the limb bud at three distinct stages, E10.5, E11.5 and E12.5, at single-cell resolution. Focusing on the limb bud mesenchyme, we have uncovered distinct transcriptional trajectories, which reveals a switch in the genetic program between E10.5 and E11.5. While at E10.5 three main transcriptional trajectories were found to coincide with distinct cell distribution along the A–P axis, at E11.5, the two main trajectories reflect differential P–D identity. Interestingly, the trajectories at E10.5 all end with cells expressing either proximal or distal markers suggesting a progressive acquisition of P–D identity. This finding is consistent with previous models suggesting that FGF signaling from the AER and proximal signaling from the flank create opposite signaling, which progressively results in the emergence of an intermediate identity separating proximal identity (stylopod) from distal identity (autopod) [1]. Our transcriptome analysis at E11.5 and E12.5 also identified three main categories of genes expressed in the distal limb mesenchyme characterized by distinct temporal expression dynamics. As anticipated from previous results, HOX13 binding was observed within or in the neighborhood of several of these genes, consistent with previous evidence suggesting that the transition from the early/proximal to the late/distal transcriptome of the limb mesenchyme largely relies on HOX13 function [11]. Finally, our data provide evidence that up to E12.5, the large majority of distal cells express the *Hox13* genes while the other distal cells are cells in which *Hox13* expression has been switched off and which undergo chondrogenic differentiation.

## 2. Materials and Methods

### 2.1. Study Approval

All studies with mice described in this article were approved by the Animal Care Committee of the Institut de Recherches Cliniques de Montréal (protocols # 2015-14 and 2017-10).

### 2.2. Mouse Lines

The generation of *Hoxa13:Cre* and *mT/mG* mouse lines was described elsewhere [12,13]. All mice were maintained in mixed background (C57BL/6X129). Noon of the day of the vaginal plug was considered as E0.5. Mice and embryos were genotyped by polymerase chain reaction (PCR) using genomic DNA extracted from tail biopsy specimens and yolk sacs, respectively.

### 2.3. 3′End Single-Cell RNA-seq

Dissection of forelimb buds from *Hoxa13:Cre/+;mT-mG* embryos was performed at E10.5, E11.5 and E12.5. Forelimbs were dissected in cold 1XPBS, and after centrifugation at 300 rcf for 5 min at 4 °C, the forelimbs were incubated in dissociation buffer (450 μL of 0.25% Trypsin/EDTA (GIBCO), 50 μL 10% bovine serum albumin (BSA), 1 μL DNAseI (NEB)) for 10 min at 37 °C. After incubation, limb cells were gently mixed by pipetting up and down 10–15 times until they were dissociated, and then 10% final fetal bovine serum (FBS) was added. Dissociated cells were filtered using a cell strainer (40 μm Nylon, BD Falcon), and counted manually using a hemocytometer to determine the final volume in which to resuspend the cells and assessed for cell viability using 0.4% Trypan blue. After centrifugation at 300 rcf for 7 min at 4 °C, the resulting pellet of cells was resuspended in the appropriate volume of 1XPBS, 0.04% BSA in order to achieve a concentration of 1500 cells/μL. Before being processed, dissociated cells were counted one more time and assessed for cell viability using 0.4% Trypan blue. The single-cell preparation was processed using Chromium Next GEM Chip G Single Cell Kit (10X Genomics PN 1000127) and Chromium Next GEM Single Cell 3′ GEM, Library and Gel Bead Kit v3.1 (10X Genomics PN 1000128) following the manufacturer’s recommendations. Briefly, forelimb cells were partitioned into gel beads in emulsion for cell lysis, barcoding with oligo-dT primers and reverse transcription in order to produce barcoded, full-length cDNA from poly-adenylated mRNA. cDNA library was amplified, fragmented and size selected. Samples were controlled at multiple steps during the procedure by running on BioAnalyzer. Libraries were sequenced on NovaSeq 6000 PE 26x98 (98-8-26).

### 2.4. Single-Cell RNA-seq Data Analysis

#### 2.4.1. Pre-Processing

The Cell Ranger Single-Cell Software Suite version3.0.1 (10X Genomics^®^) was used to perform sample demultiplexing, alignment, filtering, barcode counting, and UMI counting. Sequencing reads were aligned on a custom genome on which sequences of IRES-venus, tdTomato, Cre and eGFP were added to Mus_musculus.GRCm38.93. genome. At E10.5, 6402 cells were profiled, 6748 cells at E11.5 and 8592 cells at E12.5. Feature-barcode matrices were generated for each sample. Further analysis—including quality filtering, data normalization and scaling, identification of highly variable genes, dimensionality reduction, standard unsupervised clustering algorithms and the discovery of differentially expressed genes—was performed using the Seurat R package v3.1 [14,15].

#### 2.4.2. Quality Control and Filtering

To exclude low-quality cells, we removed cells that had unique feature counts less than 1500 or over 7500. Our samples presented a number of unique transcripts per cell that were depending on the number of total reads detected per cell following a logarithmic function that can be presented as following: $\;Unique\;Transcripts\;=\;\alpha \;\log_{2.05}\left(\#\;total\;transcript\right)$ or ${\left(Unique\;Transcripts\right)}^{2.05}\;=\;b\left(\#\;total\;transcript\right)$. We considered cells presenting a beta inferior to 800 while having a total number of transcripts superior to 3000 as doublets and filtered them out. We also removed cells with more than 12.5% and less than 2% of the transcripts coming from mitochondrial genes. After quality filtering, the number of cells per sample was 6008 cells at E10.5, 5955 cells at E11.5 and 6972 cells at E12.5.

#### 2.4.3. Normalization and Scaling

For the rest of the analysis, we analyzed each sample individually, but we also combined the three samples (E10.5, E11.5 and E12.5) using the merge function. The four Seurat objects were analyzed with the same parameters. We first normalized the data. Next, the data were scaled and during this step the mitochondrial contamination as well as cell cycle phases heterogeneity were ‘regressed out’ using the var.to.regress function in order to remove these sources of variation from our single-cell dataset. We removed cell cycle stages variation as we observed that without this step the influence of the cell cycle was taking over the cell identity variation during the clustering and was affected the downstream analysis to identify gene marker specific to a cell population. We used the CellCycleScoring function to assign cell cycle scores to each cell to do the scaling on this parameter. This scoring was based on the expression of cell cycle markers divided into G2/M phase markers, S phase markers and cells that did not express any of these markers were identified as being in G1 phase.

#### 2.4.4. Dimensional Reduction

Using the FindVariableFeatures function, 3000 features were identified as highly variable from cell to cell. Next, the lowest (mean expression < 5th percentile) and the highest (mean expression > 80th percentile) expressed genes for each dataset were removed from the 3000 highly variable features list in order to improve the downstream analysis [16]. We then performed linear dimensional reduction using the Principal Component Analysis (PCA) method including 50 Principal Components (PCs) and to visualize the data we used the Uniform Approximation and Projection method (UMAP) [17].

#### 2.4.5. Clustering

The clustering was performed using the Louvain algorithm [18,19] with a resolution of 0.3. We used the subset function in order to create a new Seurat object containing only the mesenchymal cells clusters, identify previously, for E10.5, E11.5 and E12.5. Each new Seurat object was scaled, highly variable features were identified and linear (PCA) [20] and non-linear (UMAP) dimensional reductions. From these subsets, we used the WhichCells function of Seurat [14,15] to select the *Hoxa13* positive cells in the mesenchyme and we then used the function FindAllMarkers to identify the marker genes for the *Hoxa13* positive mesenchymal cells and for the *Hoxa13* negative mesenchymal cells. From the mesenchymal subsets at E10.5, E11.5 and E12.5 we used the WhichCells function to select the *Hoxa13+;Hoxd13+;Gfp+* and the *Hoxa13−;Hoxd13−;Gfp+* cells in the mesenchyme. We then created subsets, using the subset function, composed of the selected cells and we merged them to create a new Seurat object containing the *Hoxa13+;Hoxd13+;Gfp+* (*Hox13+;Gfp+* hereafter) cells and the *Hoxa13−;Hoxd13−;Gfp+* (*Hox13−;Gfp+* hereafter) cells for each genotype. The new Seurat object was scaled, highly variable features were identified and linear (PCA) [20] and non-linear (UMAP) dimensional reductions were performed. We then used the function FindAllMarkers (min.pct = 0 and logfc.threshold = 0.25) to identify the marker genes for the *Hox13+;Gfp+* and for the *Hox13−;Gfp+* mesenchymal cells. We used DoHeatmap function to visualise the significant (p_val_adj < 0.05) marker genes for each cell population. We used these marker genes to perform the gene ontology analysis using DAVID [21,22].

#### 2.4.6. Monocle3 Analysis

We started the Monocle3 [23,24,25] analysis from the merged Seurat object (E10.5, E11.5 and E12.5), which was previously normalized, scaled with the removal of cell cycle and mitochondrial contamination variations, high variable features were identified with the removal of the highest and the lowest expressed genes and linear dimension was performed using PCA (see the Normalization and Scaling, and Dimensional Reduction sections). The function as.cell_data_set from the seurat-wrappers package was used to create the Monocle3 cds object for further analysis. We performed non-linear (UMAP) dimensional reductions, we calculated the size factors to evaluate variation in the number of mRNAs per cell, we then clustered the cells using the Leiden algorithm [26] with a resolution of 0.0003. Marker genes of each cluster were identified using the top_markers function. To analyze only the mesenchymal cells, we used the function choose_cells to subset the mesenchyme from E10.5, E11.5 and E12.5 samples, as well as E10.5 only and E11.5/E12.5 only. We clustered the subset objects using the Leiden algorithm and we used the function learn_graph to learn a principal graph in order to identify trajectories in function of the gene expression changes.

### 2.5. Single-Cell ATAC-seq Data Analysis

The single-cell ATAC-seq raw data were published [27] and can be found on GEO under the accession number GSE145657. Sequencing reads were aligned using Cell Ranger ATAC version1.1.0 (10X Genomics^®^) and Signac version1.0.0 (extension of Seurat [14,15], developed by Tim Stuart and Avi Srivastava) was used to perform the analysis. We used the function cellranger-atac count from Cell Ranger ATAC version1.1.0 (10X Genomics^®^) to perform read filtering and alignment on mm10, barcode counting, identification of transposase cut sites, detection of accessible chromatin peaks, cell calling and count matrix generation for peaks with the following parameters for the peak calling, window_size = 250 and peak_merge_distance = 5. We then used Signac version1.0.0 to analyze the Cell Ranger ATAC results. First, we created a Seurat object from the Cell Ranger ATAC files (filtered_peak_bc_matrix.h5 as the peak-feature-barcode matrices, singlecell.csv that contains per barcode information and fragments.tsv.gz that represents the list of unique fragments across all single cells). Next, we annotated using the UCSC style and we cleaned the data by eliminating cells which have a high ratio of mononucleosomal to nucleosome-free fragment sizes (i.e., >2.5), a low transcriptional start site (TSS) enrichment (i.e., <2), a very low and very high level of reads measured by the total number of fragments in peaks (i.e., <2000 and >25,000), a low fraction of fragments in peaks (i.e., <15) and reads overlapping with ENCODE Blacklist regions (i.e., >0.05). Normalization and linear dimensional reduction were performed using the latent semantic indexing (LSI) [28]. We then clustered the cells using the SLM algorithm at a resolution of 0.3 [29] and we visualized the data using UMAP [17] and the frequency of Tn5 integration as accessibility tracks.

### 2.6. Whole-Mount In Situ Hybridization

Whole-mount in situ hybridization was performed using standard procedure [27]. Briefly, embryos were rehydrated through a methanol series (100–30%), washed in PBST (0.1% Tween) and bleached for an hour on ice in the dark using 6% hydrogen peroxide. Then, embryos were treated with Proteinase K at RT for 10 min for E10.5 embryos and for 15 min for E11.5 and E12.5 embryos. Following the Proteinase K treatment embryos were re-fixed with 4% paraformaldehyde (PFA). Next, embryos were hybridized with the Digoxigenin (DIG)-labelled riboprobes in hybridization buffer (5× SSC pH 4.5; 50% deionized formamide; 1% SDS; 0.1% Tween; 5 mg/mL torula RNA, 0.5 mg/mL heparin) overnight at 68 °C. Embryos were then washed with 1× TBS; 0.1% Tween, treated with 10% goat serum; 1% BSA and incubated with alkaline phosphatase-conjugated anti-DIG antibodies (1/3000; Roche) overnight at 4 °C. The coloration was achieved using nitrotetrazolium blue chloride (NBT)/ 5-bromo-4-chloro-3-indolyl-phosphate, 4-toluidine salt (BCIP) substrate (Roche). After staining, the samples were washed in PBS and post-fixed with 4% PFA. Embryos were photographed using the Leica, Wetzlar, Germany M165FC stereomicroscope coupled to the DFC450.C camera. A minimum of three embryos per genotype was assayed for reproducibility (*n* = 3). Digoxigenin (DIG)-labelled antisense riboprobes were generated from cDNA using the following primers with the reverse primers containing the T7 promoter sequence: *Efna1*, forward: GTGACTGTCAATGGCAAAAT and reverse: TAATACGACTCACTATAGGGGAAATCTTGCAGAGATGCTG; *Creb5*, forward: CCACCCTCAGTCAGCTTACA and reverse: TAATACGACTCACTATAGGGATCATGAGCTTTCCCACCCA; *Cpa2*, forward: TGTGTTCTCCCAAACCTCCA and reverse: TAATACGACTCACTATAGGGCACTGGCCTGGTAGATGACA; *Cdh3*, forward: TGCTGACTAGGGGGACAGTT and reverse: TAATACGACTCACTATAGGGCCCTCTCCATCCATGTCTGT. *Hoxa13* and *Bmp2* probes were previously described [30,31].

### 2.7. Data Availability

The scRNA-seq raw and processed data can be found on GEO under the accession number GSE158820. The scATAC-seq raw data were obtained from the accession numbers GSE145657. The ChIP-seq data for HOXA13, HOXD13 and H3K27ac are available on GEO under the accession number GSE81356. This study did not generate any unique code, and all analyses were performed in R using standard protocols from Seurat and Monocle3 [14,15,25]. All scripts associated with this manuscript are available upon request.

## 3. Results

### 3.1. Transcriptional Trajectories in the Developing Forelimb Bud Reveal a Major Switch in the Developmental Program between E10.5 and E11.5

To gain a better understanding of the developmental program underlying limb patterning, we analyzed the transcriptome at single-cell resolution from mouse forelimb buds isolated from embryos at gestational day 10.5 (E10.5), E11.5 and E12.5 (Figure 1a). The dimensional reduction procedure applied to our single-cell RNA-seq (scRNA-seq) dataset was based on the identification of the 3000 most variable expressed genes, from which the lowest and highest expressed genes were removed. The resulting gene set was then used to conduct a Principal Component Analysis (PCA) for linear dimensional reduction (see Materials and Methods for details). To visualize the data, we then used the Uniform Approximation and Projection method (UMAP) that positions cells according to their transcriptome, with each dot representing a cell and neighboring dots corresponding to cells with the highest similarity of their transcriptome (Figure 1b). UMAP projection of our scRNA-seq dataset revealed that the majority of E10.5 cells are isolated from E11.5 and E12.5 cells (Figure 1b), suggesting a major transcriptional switch between E10.5 and E11.5. Cell clustering was then used to group cells according to similarities in gene expression, which showed a clear segregation between the main cell types, i.e., mesenchyme, ectoderm, myoblasts/myocytes, blood cells (Figure 1c,d and Appendix A), validating the biological relevance of our dataset. The cell clustering revealed that the major transcriptional switch between E10.5 and E11.5 occurred within the mesenchymal cell population (Figure 1c,d and Appendix A).

We thus focused subsequent analyses on the limb mesenchyme (Figure 2). A tentative identity for each cluster was established based on marker genes (Figure 1d and Figure 2a). In order to determine gene expression changes between developmental programs, we first established trajectories using Monocle 3 workflow [23,24,25] (Figure 2a). A trajectory establishes relationships between cells according to their transcriptome such that the transcriptome of cells that belong to two distinct trajectories are more divergent than between cells that belong to the same trajectory. At E10.5, three discrete trajectories can be identified (Figure 2b). We first identified gene markers for each trajectory. The first trajectory is characterized by cells expressing *Irx3*, which marks the anterior cell population [32]. The second one is marked by *Grem1* expression, which characterizes the central domain of the limb bud [33] and the third trajectory includes cells expressing *Shh*, located at the posterior margin of the developing limb bud [34]. These well-characterized markers allowed us to orient the UMAP projection and position the trajectories in the mouse limb bud (Figure 2b). We then used other well-known markers to validate the differential antero–posterior identity of the trajectories. *Alx4*, which is expressed in the anterior limb bud [35], was found in trajectory 1. *Tbx2*, expressed both in the anterior and posterior margins [36], was present in trajectories 1 and 3 and excluded from trajectory 2 while *Hoxd12*, expressed in the posterior half domain [37], was found in trajectories 2 and 3 and excluded from trajectory 1. In contrast to genes having a differential antero–posterior expression, *Meis1*, which characterizes the proximal limb domain [38], was observed in the three trajectories. Of note, *Hoxa13*, which is expressed distally with a posterior bias [39,40], was present at the extremity of trajectory 2 and 3. Together, the distribution of these marker genes on the UMAP projection indicated that trajectory 1 corresponds to anterior limb cells, trajectory 2 to the central limb domain and trajectory 3 to the posterior limb (Figure 2b). These data thus suggested that antero–posterior patterning is the primary patterning event in early limb buds.

As far as the E11.5 and E12.5 cell population is concerned, we found two main trajectories, which split into ‘secondary’ branches. Of note, the extremity of almost all ‘branches’ is composed of E12.5 cells (Figure 2c). The marked segregation between E10.5 and E11.5 clustering prevented us from connecting the E10.5 trajectories with those of E11.5 and E12.5 limb bud cells. Nonetheless, based on the genes characterizing each trajectory, a clear switch could be observed whereby the main trajectories at E10.5 coincided with distinct A–P distribution while the two main trajectories at E11.5 and E12.5 limb buds correspond to a differential P–D identity as revealed by the distribution of cells expressing the proximal markers *Shox2* and *Hoxa11* [41,42,43] and cells expressing the distal markers *Hoxa13* and *Hoxd13* [39,40](Figure 2c).

### 3.2. Individual Transcriptional Trajectories Are Characterized by a Specific Hox Code

We next examined the distribution of *Hox*-expressing cells to further analyze the trajectories in the context of limb bud patterning into three distinct domains along the P–D axis—the stylopod, zeugopod and autopod. We chose the *Hox* genes for their well-characterized differential expression along both the P–D and A–P axes and their known impact on the limb architecture (reviewed in, e.g., [4]). At E10.5, cells expressing *Hoxa9, d9, a10* and *d10*, which pattern the stylopod, are present in all three trajectories (Figure 3a). *Hoxa11* and *Hoxd11*, responsible for the zeugopod formation, are expressed in cells belonging to trajectories 2 and 3, which also express *Hoxd12* but to a lesser extent. Finally, cells expressing *Hoxd13* are observed in trajectory 3 and the extremity of trajectory 2 while cells expressing *Hoxa13* are only found at the extremity of trajectories 2 and 3 (Figure 3a). Together, these data revealed that each trajectory is characterized by the expression of a specific combination of *Hox* genes, which in turn further support the differential A–P identity of the three transcriptional trajectories identified at E10.5. For limb tissue at E11.5 and 12.5, the dissection plan did not include the most proximal cells to avoid ‘contamination’ from flank tissue (Figure 1a). As a result, rare *Meis1/2*-expressing cells were observed in the E11.5 and E12.5 samples (Appendix A). This together with the relative prevalence of the distal limb tissue as compared to the proximal tissue explains the large proportion of E11.5 and E12.5 mesenchymal cells expressing *Hoxa13* and *Hoxd13* (Appendix A). We also found that *Hoxa9, Hoxa10, Hoxd10* and *Hoxd11* are expressed in virtually all limb bud cells (Figure 3b). *Hoxa11* and *Hoxa13* have an almost mutually exclusive distribution (Figure 3c), as expected from previous whole mount in situ hybridization data [43,44,45] and consistent with the HOX13-mediated repression of *Hoxa11* [43]. Of note, there are few cells that express both *Hoxa11* and *Hoxa13*, which possibly correlate with cells in which *Hoxa13* is transcribed but the HOXA13 protein is not yet produced/functional (Figure 3c, red arrows). These cells also express the proximal marker *Shox2* (Figure 3c, red arrows), suggesting that these cells may not have yet acquired a definitive P–D identity. We also observed that *Hoxd9* is expressed in all limb bud cells at E10.5 but is barely expressed in distal cells at E11.5 and E12.5, similar to *Hoxa11* (Figure 3c, purple arrows). Together, these data highlight the fact that, at E11.5–E12.5, only five of the *Hox* genes have a spatially restricted expression, namely *Hoxd9*, *Hoxa11, Hoxd12, Hoxa13* and *Hoxd13* (Figure 3c), while the others are expressed in virtually all cells though with distinct expression levels along the P–D axis (Figure 3b).

### 3.3. Dynamic Expression of Distal Genes Largely Relies on HOX13 Function

We next focused our analysis on distal mesenchymal cells. We used *Hoxa13* (Figure 4a), which is expressed in the progenitor cells of the hand and wrist, as a marker to identify distal mesenchymal cells [12]. Analysis of the transcriptome of distal cells at E10.5, E11.5 and E12.5 revealed three categories of ‘distal’ genes based on the temporal dynamics of their expression (Figure 4b and Appendix A). Category 1 corresponds to genes not expressed at E10.5 but becoming expressed in *Hoxa13*-positive cells at E11.5 (e.g., *Creb5, Bmp2*, *Efna1,* and *Stmn2*). Category 2 represents genes whose transcription is found among the distal cells at E10.5 and E11.5 while only rare cells at E12.5 express these genes (e.g., *Gja3, Cpa2,* and *Cbfa2t3*). Finally, Category 3 genes are found expressed over the three stages in a subset of distal cells (e.g., *Cdh3*, *Lmo2*, *Tfap2b, Jag1,* and *Hey1*).

To gain insights as to whether these genes are regulated by HOX13 TFs, we checked whether HOXA13 and/or HOXD13 proteins bind in their vicinity and analyzed chromatin accessibility at single-cell resolution (scATAC-seq). HOXA13, HOXD13 and H3K27ac ChIP-seq data as well as the scATAC-seq data were those previously generated from E11.5 limb buds [11,27]. Analysis of scATAC-seq and ChIP-seq data reveals that genes from all three categories are in an accessible state and decorated with the H3K27ac active mark in the entire limb bud (Figure 5a and Appendix A), suggesting that their distal expression is due to transcriptional enhancers specifically active in the distal mesenchyme. Interestingly, we found that genes from all three categories are associated with HOX13 ChIP-seq peaks, either within the gene (e.g., *Creb5*), downstream (e.g., *Gja3* and *Efna1*), or upstream (e.g., *Bmp2* and *Tfap2b*). Loci bound by HOX13 are characterized by a cell-specific accessibility (Figure 5b). Notably, the HOX13-bound loci are all in an inaccessible/reduced accessibility in the ‘proximal’ and ‘middle 1’ clusters, while they are all in an accessible state in the ‘distal’ clusters. Together with the evidence that these loci are decorated with the active H3K27ac histone mark (Figure 5b), these data suggest that the HOX13-bound loci correspond to distal-specific regulatory elements. The HOX13-bound loci are also in an accessible state in the cluster referred to as ‘middle 2’, which may coincide with cells expressing both *Hox13* and proximal markers, as described above, which possibly corresponds to cells that have not yet acquired their definitive P–D identity. Although our data do not demonstrate whether the HOX13-bound loci reported here are the bona fide regulatory elements controlling these distal genes, the evidence that these loci are in an accessible state and coincide with regions decorated by the active H3K27ac histone mark (Figure 5b) supports a direct contribution of HOX13 in the regulation of this set of distal genes.

While *Hoxa13* and *Hoxd13* establish the distal identity in the developing limb bud, how the naturally occurring termination of their expression impacts the fate of distal limb cells remains elusive. To address this question, our scRNA-seq analysis was performed using limb buds from the *Hoxa13Cre/+; mT/mG* line, which allows the identification of cells originating from *Hoxa13*-expressing cells, including those that do not express *Hoxa13* anymore (*Hoxa13−;Gfp+* cell population in our scRNA-seq data). Previous analysis of the *Hoxa13Cre/+; mT/mG* mice provided evidence that, in the forelimb mesenchyme, all *Hoxa13*-expressing cells are progenitors of the hand and wrist [12], i.e., all *Gfp*-expressing mesenchymal cells in *Hoxa13Cre/+; mT/mG* limb buds are cells with distal identity. We thus compared the transcriptome of *Gfp*-expressing cells which express neither *Hoxa13* nor *Hoxd13* (*Hox13−;Gfp+*) and those expressing *Gfp, Hoxa13* and *Hoxd13* (*Hox13+; Gfp+*). At E10.5, we identified only five *Hox13−;Gfp+* cells while 108 and 309 *Hox13−;Gfp+* cells were found at E11.5 and E12.5 respectively (Appendix A). We thus focused our analysis on the E11.5 and E12.5 datasets. Transcriptome comparison between *Hox13−;Gfp+* and *Hox13+;Gfp+* cells revealed that the termination of *Hoxa13* and *Hoxd13* expression in distal cells is primarily associated with the upregulation of chondrogenic markers (Appendix A). This indicated that the distal limb cells in which *Hoxa13* and *Hoxd13* expression is switched off are, at least up to E12.5, distal cells undergoing chondrogenic differentiation.

## 4. Discussion

In an attempt to gain further insights into the developmental program underlying limb patterning, we have performed and analyzed scRNA-seq experiments using isolated mouse forelimb buds from three stages, namely E10.5, E11.5 and E12.5. Analysis of this scRNA-seq dataset revealed a number of features characterizing the limb developmental program, including features previously reported [1,9,11] which validate our experimental approach.

Our analysis highlights a major transition in gene expression between E10.5 and E11.5 and suggests a switch from a prevalent A–P patterning program in the early limb bud to a prevalent P–D program at E11.5–E12.5. Indeed, while the main transcriptional trajectories at E10.5 are associated with distinct A–P markers, at E11.5–E12.5 there is a switch from three to two main trajectories, which are characterized by the expression of respectively proximal and distal markers. Next, we have investigated the distribution of cells expressing *Hox* genes, which are known for their critical function in establishing the architecture of the limb skeleton (e.g., [4]). At E10.5, the distribution of the *Hox* transcripts varies between the three identified trajectories according to their differential A–P expression, further supporting that at E10.5, each trajectory corresponds to a distinct limb bud domain along the A–P axis. At E11.5 and E12.5, based on our scRNA-seq dataset, *Hox* genes expressed in limb buds can be categorized into two groups: those expressed in virtually all cells, though with distinct expression levels (*Hoxa9, a10, d10* and *d11*) and those only expressed in a subset of limb cells (*Hoxd9*, *Hoxa11, Hoxd12, Hoxd13* and *Hoxa13*). Previous work provided evidence that the proximal restriction of *Hoxa11* expression is the consequence of HOX13-mediated regulation, whereby HOX13 triggers antisense transcription at the *Hoxa11* locus, which, as a result, prevents *Hoxa11* expression in the distal limb bud [43]. In this view, the key event that underlies *Hox* differential expression along the P–D axis of the developing limb, at least the qualitative aspect of it, is the regulatory mechanism responsible for *Hox13* transcriptional activation being restricted to the distal domain of the developing limb bud. This model is consistent with the evidence that the *Hox13* function is required for the silencing of the regulatory landscape responsible for the proximal limb expression of the *HoxD* genes [10,11]. Interestingly, we found that a subset of cells expresses both *Hox13* and *Hoxa11*, raising the possibility that these cells coincide with cells in which the *Hox13* genes are expressed but the HOX13 proteins are either not yet produced or not yet functional. These cells also express the proximal maker *Shox2* suggesting that they have not yet acquired a definitive P–D identity. These cells could correspond to the progenitors of the wrist, located in between the zeugopod and the digits. This latter view is consistent with previous evidence showing that *Hoxa13*-expressing cells are progenitor cells of both the digits and wrist [12].

Based on the importance of *Hox13* distal-specific expression in establishing distal limb identity, we have further focused the analysis of our scRNA-seq dataset on genes expressed in *Hox13*-expressing cells. We found three categories of genes based on the developmental dynamics of their expression. Analysis of randomly selected genes from each category revealed HOX13 binding within the genes or within a 100kb window around the genes. Based on the evidence that these HOX13-bound loci are in an accessible state specifically in distal limb cells and are marked by H3K27ac, which reflects a transcriptionally active status, it is likely that the majority of distally expressed genes are direct targets of HOX13. Accordingly, previous work provided evidence that the activity of distal-specific regulatory modules largely relies on HOX13 function [11,27]. These results, together with the evidence that an overall transition from A–P to P–D patterning occurs concomitantly with the beginning of *Hox13* expression, support a model whereby the HOX13 transcription factors have a prevalent role in the P–D patterning process.

Among the three categories of genes expressed in the distal limb bud, Category 1 are genes that become expressed at E11.5 and include genes whose activation in distal cells relies on HOX13 pioneer activity, as exemplified with *Bmp2* [27]. Category 2 and 3 genes are those already expressed at E10.5 and thereby include upstream regulator(s) of the *Hox13* genes as well as genes regulated by HOX13. Category 2 genes differ from Category 3 by being downregulated at E12.5, suggesting that HOX13 are not the primary factors controlling the expression of Category 2 genes, at least after E11.5. This category includes *Gja3*, known to be expressed in human chondrocytes [46] and *Cpa2* [25]. Finally, Category 3 represents genes expressed at the three stages examined and include *Tfap2b*, whose function is associated with digit formation [47].

The last aspect that we investigated in this study was to establish which cell fate(s) is associated with the termination of *Hox13* expression. Taking advantage of a genetic lineage tracing tool, we were able to analyze the transcriptome of distal cells in which *Hoxa13* and *Hoxd13* are not expressed anymore. At least up to E12.5, these cells represent a small fraction of distal cells and are cells undergoing chondrogenic differentiation. However, it remains to be established whether chondrogenic differentiation can only be implemented upon the termination of *Hoxa13* and *Hoxd13* expression or is part of the mechanism that switch-off both genes.

## 5. Conclusions

In summary, the present analysis of scRNA-seq data from mouse forelimb buds from E10.5 to E12.5 highlights a major transcriptional transition between E10.5 and E11.5, with transcriptional trajectories switching from A–P to P–D patterning programs. This study also identifies a subpopulation of limb mesenchymal cells, which have either not yet resolved their P–D identity at E12.5 or have an intermediate identity. These cells could correspond to progenitor cells of the wrist, which start forming at a stage when digit and zeugopod anlagen are clearly distinguishable. Finally, our scRNA-seq data identify three categories of genes expressed in *Hox13*-expressing cells, all three including genes whose transcriptional control involves HOX13 function. This later dataset should be useful for future studies aimed at unraveling the genetic network, triggering the emergence of digits as well as identifying transcription factors responsible for the distal restriction of the HOX13 pioneer factors.

## Figures and Tables

**Figure 1 jdb-08-00031-f001:**
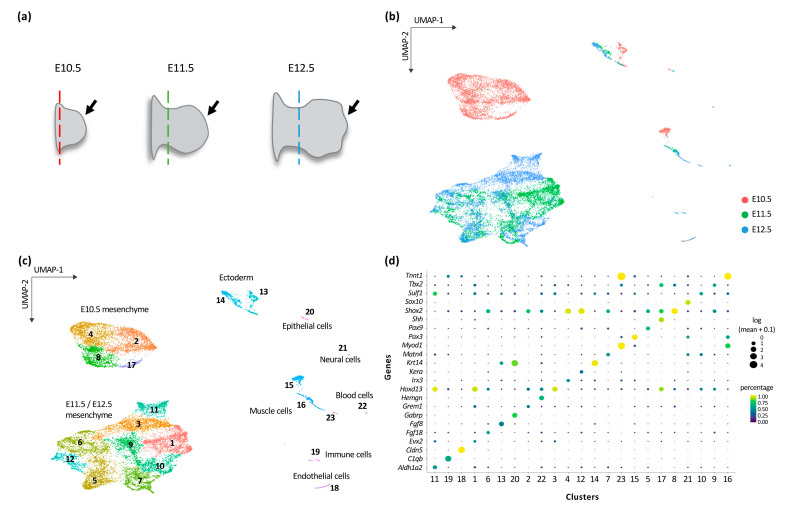
Identification of the different cell identity in the developing mouse limb bud. (**a**) Schematics illustrating mouse forelimb buds tissues used in this study: E10.5 (left, black arrow), E11.5 (middle, black arrow) and E12.5 (right, black arrow). The dashed color lines mark the position of the dissection. (**b**,**c**) UMAP visualization of 18,935 mouse forelimb buds cells, colored by developmental stages (6008 cells for E10.5 in red; 5955 cells for E11.5 in green and 6972 cells for E12,5 in blue) (**b**) and by cluster identity from Leiden clustering (**c**). The clusters were annotated based on the identification of marker genes. (**d**) Dot plot showing the expression of one selected marker gene per cluster. The size of the dot represents the mean of the expression level of the gene and its color represents the percentage of cells within the cluster in which that gene was detected.

**Figure 2 jdb-08-00031-f002:**
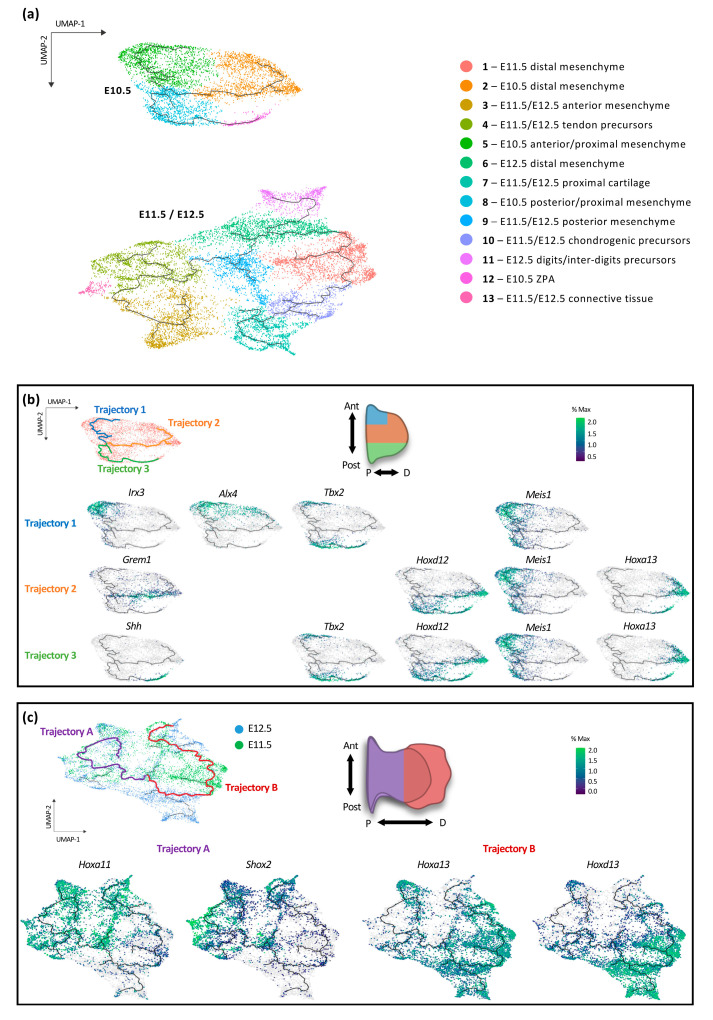
Characterization of the main transcriptional trajectories in the developing limb mesenchyme. (**a**) UMAP visualization of the forelimb bud mesenchymal cells colored according to cluster identity on which developmental trajectories, identified using Monocle 3, are represented. (**b**) Characterization of the three mesenchymal trajectories at E10.5, trajectory 1 is highlighted in blue, trajectory 2 in orange and trajectory 3 in green on the UMAP representation (top left). The E10.5 forelimb bud schematic indicates the orientation of the UMAP representation and the localization of the trajectories (middle top). Expression profiles of marker genes for each trajectory are shown below. Color scale indicated the scaled expression to percent of the maximum expression of all marker genes at E10.5 (top right). (**c**) Identification of the two main transcriptional trajectories at E11.5 (green cells) and E12.5 (blue cells), trajectory A is highlighted in purple and trajectory B in red (top left). E11.5 and E12.5 overlapping limb bud schematics indicate the orientation of the UMAP representation and the localization of the trajectories (middle top). Expression profiles of marker genes for each trajectory are shown below (bottom left and right, respectively). Color scale indicated the scaled expression to percent of the maximum expression of all marker genes at E11.5 and E12.5 (top right). Ant: anterior, Post: posterior, P: proximal, D: distal.

**Figure 3 jdb-08-00031-f003:**
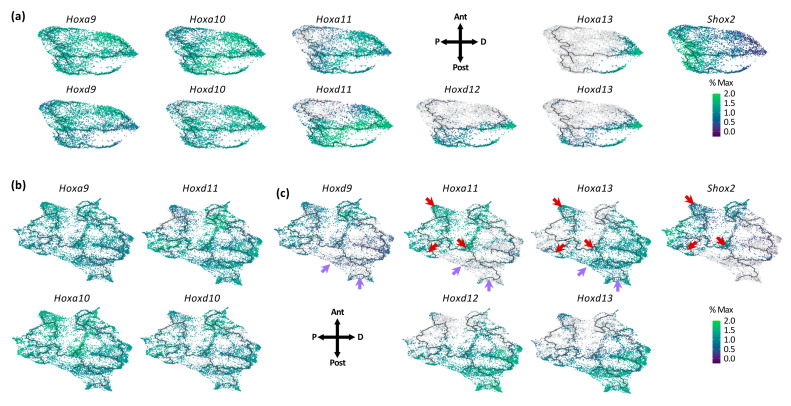
Distribution of *Hox*-expressing cells along the transcriptional trajectories during limb bud development. (**a**) UMAP visualization of the mesenchymal cells expressing the most 5’ *Hoxa* and *Hoxd* genes (*Hoxa9*, *Hoxd9*, *Hoxa10*, *Hoxd10*, *Hoxa11*, *Hoxd11*, *Hoxd12*, *Hoxa13* and *Hoxd13*) as well as the cells expressing *Shox2* at E10.5. Color scale indicated the scaled expression to percent of the maximum expression of all genes at E10.5. Orientation of the UMAP representation is indicated by the cross arrow schematic. (**b**,**c**) UMAP visualization of E11.5 and E12.5 mesenchymal cells expressing *Hoxa9*, *Hoxa10*, *Hoxd10* and *Hoxd11* in the all mesenchyme (**b**) and expressing *Hoxd9*, *Hoxa11*, *Hoxd12*, *Hoxa13, Hoxd13* and *Shox2* in restricted areas of the mesenchyme (**c**). Red arrows indicate co-expression areas of *Hoxa11*, *Hoxa13* and *Shox2*, and light purple arrows show the areas in which *Hoxd9* and *Hoxa11* are not expressed but *Hoxa13* is expressed (**c**). Color scale indicated the scaled expression to percent of the maximum expression of all genes at E11.5 and E12.5. Orientation of the UMAP representation is indicated by the cross arrow schematic. Ant: anterior, Post: posterior, P: proximal, and D: distal.

**Figure 4 jdb-08-00031-f004:**
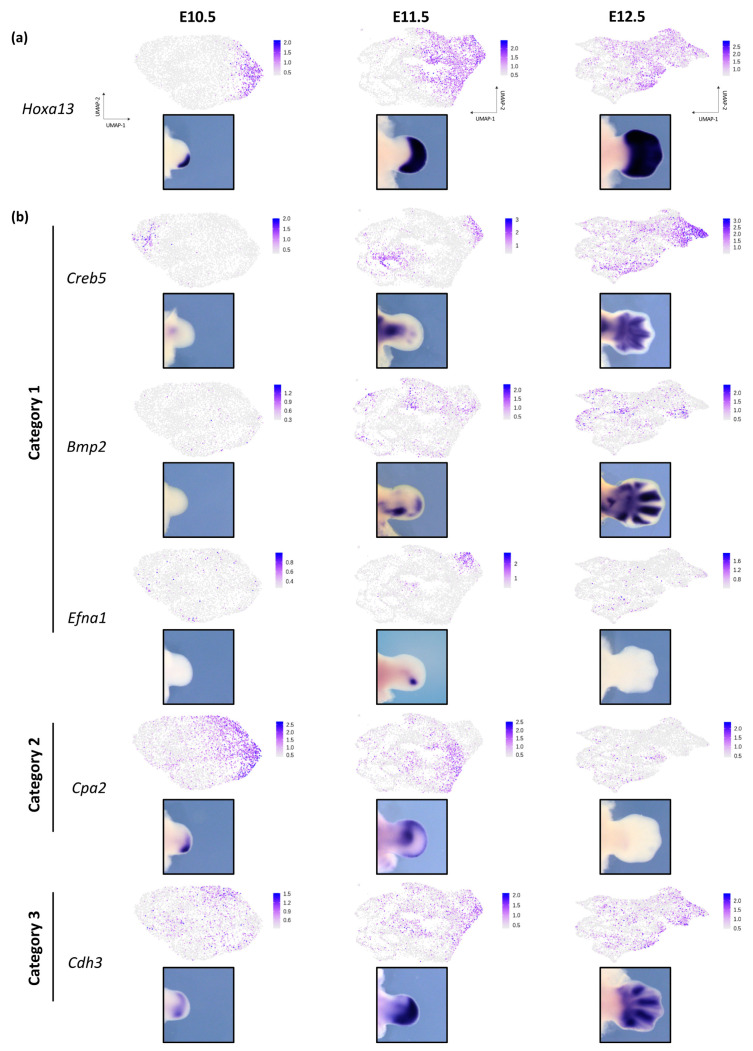
Dynamic expression of ‘distal’ genes during limb bud development. (**a**) UMAP visualization of the mesenchymal cells expressing *Hoxa13* and whole-mount in situ hybridization for *Hoxa13* in WT mouse forelimb bud at E10.5, E11.5 and E12.5. (**b**) UMAP visualization and whole-mount in situ hybridization of the distal genes *Creb5, Bmp2*, *Efna1* belonging to Category 1, *Cpa2* belonging to category2 and *Cdh3* belonging to Category 3, in E10.5, E11.5 and E12.5 WT mouse forelimb buds. Color scale represents the scaled expression for each gene.

**Figure 5 jdb-08-00031-f005:**
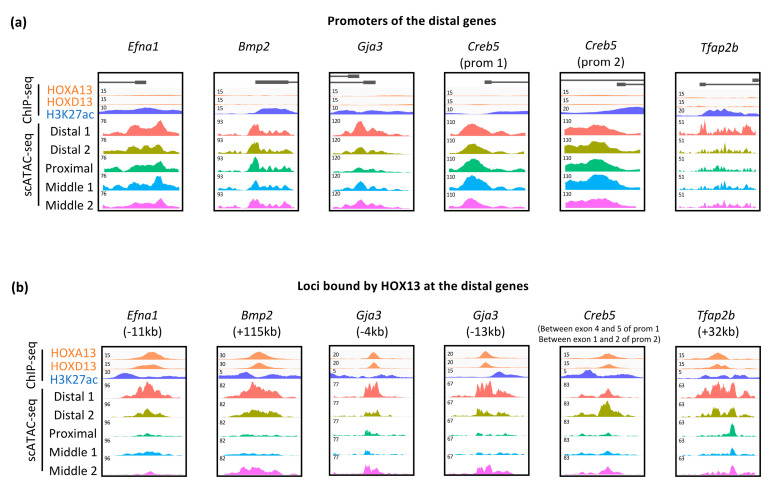
HOX13 binding at regulatory loci of distal genes correlates with distal-specific chromatin opening. (**a**,**b**) Genome views of HOXA13, HOXD13 and H3K27ac ChIP-seq (IGV) as well as scATAC-seq signals (Signac) at distal genes promoters (**a**) and at their previously identified or potential distal limb enhancers (**b**) in E11.5 mouse forelimb buds. Prom: promoter.

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
