# Peer review of "Transcriptional Trajectories in Mouse Limb Buds Reveal the Transition from Anterior-Posterior to Proximal-Distal Patterning at Early Limb Bud Stage"

_jdb, 2020, doi:10.3390/jdb8040031_

Round 1
Reviewer 1 Report
How positional fates within the developing limbs are being established has been a long-standing question. Major players in establishing the proximal-distal (PD) and the anterior-posterior (AP) axes have been identified in the past and Hox genes play important roles in these processes. This manuscript analyzes the transcriptome during 3 stages of early mouse limb development at the single cell level with a focus on the limb mesenchyme. Its findings imply that there is a global switch from AP to PD patterning mechanisms between early and later limb development and a progressive acquisition of distal identity. Finally, the authors identify Hox13-regulated gene sets that define 3 different categories of transcriptional profiles in distal mesenchymal limb bud cells.
This paper is very topical and expands - as well as confirms - the knowledge gained in recent years on the role of Hox genes in limb development. The quality of the data appears to be excellent, as many known observations are confirmed, hence the novel findings can be trusted, particularly the identification of 3 categories of distal cells with representative gene expression signatures.
My only major concern is that some of the technical details are not intuitively comprehensible for readers less familiar with the data analyses.
- It should be explained how the Hoxa13:Cre and mT/mG mouse lines contributed to the project (that information is only partly covered in paragraph 2.1.). The manuscript would benefit from a brief overview of which cells are labeled by Hoxa13:Cre (and when) and by information how the Cre reporter was utilized (I presume cell sorting?).
- A concise explanation of the dimensional reduction procedure and the dots in the UMAP visualizations (each a single cell sequence?) should be included for readers less familiar with the method. What does sliding along the UMAP-axes actually translate to in the expression profiles?
- How does one interpret trajectories (2b)? Single cells whose expression profile bridges that of two neighbouring ones in an effort to connect major territories? Please give a example/explanation of what a trajectory would represent.
Typos:
Line 374: Hoxa9, not Hoxd9, is expressed in all cells – change Hox9 to Hoxa9.
Fig. 5 contains errors in internal headlines, i.e. closed brackets instead of blanks and inappropriate “?” and “&”.
Author Response
First of all, we want to thank the reviewer for his/her insightful comments, which help us improve our manuscript.
Point-by-point response to the reviewer's comments is provided below (reviewer's comments are in italics).
- It should be explained how the Hoxa13:Cre and mT/mG mouse lines contributed to the project (that information is only partly covered in paragraph 2.1.). The manuscript would benefit from a brief overview of which cells are labeled by Hoxa13:Cre (and when) and by information how the Cre reporter was utilized (I presume cell sorting?).
The Hoxa13:Cre;mT/mG line allows the identification of cells that stem from Hoxa13-expressing cells, whether they still express Hoxa13 (Hoxa13+;Gfp+) or not (Hoxa13-;Gfp+). Previous analysis of the Hoxa13:Cre;mT/mG line showed that all Gfp+ mesenchymal cells are progenitor cells of the wrist and digits (Scotti et al., 2015) and thereby all Gfp+ mesenchymal cells are cells of distal identity. Our idea was to take advantage of this genetic lineage tracing tool to assess how the naturally occurring switch-off of the Hox13 genes impacts on cell fate by extracting, from our scRNA-seq data set, the transcriptome of cells expressing Gfp but neither Hoxa13 nor Hoxd13 (Hox13-;Gfp+) and compare it with the transcriptome of cells expressing Gfp, Hoxa13 and Hoxd13 (Hox13+;Gfp+). Because at the stages examined in this work, the Hox13-;Gfp+ cell population corresponds to a very small fraction of all Gfp+ cells, we had initially decided not to include this analysis. However, we agree with the reviewer that we should have explained why we have performed our scRNA-seq experiment using the Hoxa13:Cre; mT/mG mouse line. This is now explained at the end of paragraph 3.3 (lines 376 to 385). Even though the Hox13-;Gfp+ cell population corresponds to a very small fraction of all Gfp+ cells, their analysis provide information regarding the fate of the first mesenchymal cells in which Hoxa13 and Hoxd13 are naturally switched-off and therefore we have added this analysis in the revised version of our manuscript (end of paragraph 3.3, lines 385 to 392; new Figure S5 and last paragraph of the discussion section, lines 454 to 460).
- A concise explanation of the dimensional reduction procedure and the dots in the UMAP visualizations (each a single cell sequence?) should be included for readers less familiar with the method. What does sliding along the UMAP-axes actually translate to in the expression profiles?
The dimensional reduction procedure was based on the identification of the 3,000 most variable features from cell-to-cell (in our single cell data, features correspond to genes) from which the lowest and the highest expressed genes were removed. This selection of highly variable gene expression was then used in order to conduct a Principal Component Analysis (PCA) analysis. PCA is a dimensionality-reduction method that is often used to reduce the dimensionality of large data sets, by transforming a large set of variables into a smaller one that still contains most of the information of the large set. Smaller data sets are easier to explore and visualize and make data analysis much easier and faster for machine learning algorithms. In our case the highly variable gene expression identified and 50 principal components were used. Then to visualize the data we used the Uniform Approximation and Projection method (UMAP) that positions cells according to their transcriptome, where neighboring cells correspond to cells with the highest similarity of their transcriptome (in our case, in a 2-dimensional space). On the UMAP representation each dot represents a single cell. Sliding along the UMAP-axes thus translates into increased differences in the expression profile of the limb bud cells but do not give any information regarding the cell position in the tissue. However, using well-characterized markers for proximal, distal, anterior and posterior tissues, we were able to introduce positional information on the UMAP projection (Figure 2b-c).
These explanations of the dimensional reduction procedure and UMAP visualization are now added in the result section (first paragraph of section 3. Lines 247 to 254).
- How does one interpret trajectories (2b)? Single cells whose expression profile bridges that of two neighbouring ones in an effort to connect major territories? Please give a example/explanation of what a trajectory would represent.
A trajectory, identified in an unsupervised manner using Monocle 3, establishes relationships between cells according to their transcriptome. Cells that belong to two distinct trajectories are more divergent with respect to their transcriptome than cells that belong to the same trajectory. Thereby, each trajectory represents a distinct developmental program. For instance, in Figure 2, the trajectories at E10.5 highlights 3 distinct developmental programs. Based on marker genes for each trajectory, we were able to establish that these 3 developmental programs correspond respectively to the developmental program characterizing anterior, middle and posterior limb bud cells, thereby indicating that antero-posterior patterning is the primary patterning event at E10.5.
A concise explanation of transcriptional trajectories is added in the revised version of the manuscript (2nd paragraph of section 3.1; lines 266 to 268). In addition, the second paragraph of section 3.1 has been revised to include the validation that the E10.5 trajectories are transcriptional trajectories associated with A-P patterning. This validation is based on transcripts distribution of genes with well-characterized differential expression along the A-P axis (new Figure 2; 2nd paragraph of section 3.1, lines 274 to 284).
- Typos:
Line 374: Hoxa9, not Hoxd9, is expressed in all cells – _change Hox9 to Hoxa9.
Fig. 5 contains errors in internal headlines, i.e. closed brackets instead of blanks and inappropriate “?” and “&”.
Hox9 is now corrected to Hoxa9 on line 374 of the initially submitted manuscript (line 410 in revised manuscript).
We apologize for the errors contained in the Figure 5. These errors were not present on the word document that we had uploaded on the journal website and seem to be associated with the online conversion to a pdf document. We have now uploaded the figures as separate files and verified that there is no error.
Reviewer 2 Report
In this manuscript, Desanlis et al., aim to dissect mouse limb development and the underlaying gene circuits using single-cell RNA-Seq (scRNA-Seq). The authors specifically analyze the gene expression at single-cell resolution of three stages relevant for the limb development: E10.5, E11.5, and E12.5. Using off-the-shelf bioinformatic packages to analyze scRNA-Seq data, the authors determine that at E10.5, the gene expression signatures reflect the acquisition of anterior-posterior patterning. This signature is replaced in the subsequent stages by the expression of genes involved in proximal-distal patterning. Next, the authors focus on the expression of Hox genes, to find that a subset of Hoxa and Hoxd genes have precise spatial expression while the rest are expressed in all tested cells albeit with an expression gradient along the posterior-distal axis. Interestingly, scRNA-Seq is inherently blind to cell position in the original organ, but by analyzing Hox genes with well-established expression patterning, the authors are able to elegantly introduce pseudo proximal-distal and anterior-posterior coordinates in their data. Next, the authors focused on the chromatin landscape of a particular subset of cells expressing Hoxa13, which will populate the hand and wrist. The authors use publicly available ChIP-Seq and single-cell ATAC-Seq to uncover that genes associate with Hox13 maintain open chromatin configurations and are bound by Hox13 transcription factors in the entire limb bud. These results led the authors to suggest that the distally-restricted expression of these genes is controlled by distal-specific enhancers.
Albeit some of the findings in this manuscript were already reported using more classic techniques, the work by Desanlis et al. is rigorous and will provide a very good resource of gene expression during limb development at single-cell resolution. The data in this manuscript could spur future research focused in elucidating the transcription factors/enhancers that restrict the expression of Hox13 targets in the distal region of the limb. Therefore, this manuscript would be of interest to the developmental community and the readers of Journal of Developmental Biology.
Author Response
We are thankful to the reviewer for his/her comments and for supporting the publication of our manuscript in the Journal of Developmental Biology.
Reviewer 3 Report
Desanlis and colleagues described the evolution of limb using a single-cell approach. They took advantage of recent technological advancements in the scRNAseq to discover changes in gene expression of cells participating in the formation of mouse limb. The question developed in the manuscript is very interesting, but the article appears to describe a proof of concept because it is laking for validation experiments.
Major comments.
- Reading the title it is impossible to understand that the work is done with the mouse as a model organism. Please include this information in the title.
- Please include the method used to count cells. In fact, it is important for the number of cells for good library production. Did the Authors use manual or automatic cell counter?
- Authors evidenced with single-cell analysis (RNAseq) that not all Hox genes have restricted spatial expression. Authors evidenced a lot of markers. It should be interesting also a validation of what single-cell analysis allowed discussing (e.g using FISH on sequential tissue sections). Authors also evidenced co-expressed genes with Hox genes. Also this aspect could be considered in the validation steps instead of demonstrating the spatial expression of HOX genes or other markers Authors want to validate.
Minor comments.
Please control font along all the manuscript and uniform them. E.g. tilte 3.1 is in a different font from other titles of the same level.
Author Response
First of all, we would like to thank the reviewer for his/her comments to improve the quality of our manuscript.
A point-by-point response to the reviewer's comments is provided below (reviewer's comments are in italics)
- Reading the title it is impossible to understand that the work is done with the mouse as a model organism. Please include this information in the title.
The reviewer is correct and the title has been modified to include the information that the analysis was made using mouse as a model system. (Lines 2 to 5)
- Please include the method used to count cells. In fact, it is important for the number of cells for good library production. Did the Authors use manual or automatic cell counter?
We have counted cells manually using a hemocytometer. This information is now included in the method section (paragraph 2.3, line 100). After quality control and filtering (see paragraph 2.4.2), sequencing data from 6008 cells at E10.5, 5955 cells at E11.5 and 6972 cells at E12.5 were used for our analyses.
- Authors evidenced with single-cell analysis (RNAseq) that not all Hox genes have restricted spatial expression. Authors evidenced a lot of markers. It should be interesting also a validation of what single-cell analysis allowed discussing (e.g using FISH on sequential tissue sections). Authors also evidenced co-expressed genes with Hox genes. Also this aspect could be considered in the validation steps instead of demonstrating the spatial expression of HOX genes or other markers Authors want to validate.
For genes expressed in Hox13-expressing cells, we have now added whole-mount in situ hybridization of a subset of genes from each category, which validate both their distal limb expression and expression dynamics. See new Figure 4.
Other validations have been added or better explained in the revised version of the manuscript.
a. The limb bud is known to be formed by distinct cells types (e.g. muscle cells, ectoderm, mesenchyme….). In our scRNA-seq data set, each cell type can be distinguished and form its own cluster, which validates the biological relevance of our data set (Figure 2; 1st paragraph section 3.1, lines 256-259)
b. Genes with well characterized differential A-P expression pattern (as evidence by published whole-mount in situ hybridizations) are now used to validate the distinct A-P identity of each transcriptional trajectory at E10.5 (new Figure 2). We have modified the 2nd paragraph of section 3.1 to include this validation (lines 275 to 286).
c. The differential A-P identity of each trajectory at E10.5 is further validated with the differential distribution of the Hox transcripts among the trajectories (revised section 3.2, lines 321 to 323; Figure 3a).
- Minor comments. Please control font along all the manuscript and uniform them. E.g. tilte 3.1 is in a different font from other titles of the same level.
We apologize for the formatting errors. These have now been corrected throughout the text.
Round 2
Reviewer 3 Report
The Authors have answered all my questions. The manuscript can be published.